# Effects of Osthol Isolated from *Cnidium monnieri* Fruit on Urate Transporter 1

**DOI:** 10.3390/molecules23112837

**Published:** 2018-11-01

**Authors:** Yuusuke Tashiro, Ryo Sakai, Tomoko Hirose-Sugiura, Yukio Kato, Hirotaka Matsuo, Tappei Takada, Hiroshi Suzuki, Toshiaki Makino

**Affiliations:** 1Department of Pharmacognosy, Graduate School of Pharmaceutical Sciences, Nagoya City University, Nagoya 467-8603, Japan; ytashiro@sunprom.med.nagoya-cu.ac.jp (Y.T.); sakairyo0@gmail.com (R.S.); 2Faculty of Pharmacy, Institute of Medical, Pharmaceutical and Health Sciences, Kanazawa University, Kanazawa 920-1192, Japan; tomoko.s@p.kanazawa-u.ac.jp (T.H.-S.); ykato@p.kanazawa-u.ac.jp (Y.K.); 3Department of Integrative Physiology and Bio-Nano Medicine, National Defense Medical College, 3-2 Namiki, Tokorozawa, Saitama 359-8513, Japan; hmatsuo@ndmc.ac.jp; 4Department of Pharmacy, The University of Tokyo Hospital, Faculty of Medicine, The University of Tokyo, Tokyo 113-8655, Japan; tappei-tky@umin.ac.jp (T.T.); suzukihi-tky@umin.ac.jp (H.S.)

**Keywords:** urate transporter 1, uric acid, traditional Japanese Kampo medicine, *Cnidium monnieri*, osthol, coumarins

## Abstract

(1) Background: Crude drugs used in traditional Japanese Kampo medicine or folk medicine are major sources of new chemical entities for drug discovery. We screened the inhibitory potential of these crude drugs against urate transporter 1 (URAT1) to discover new drugs for hyperuricemia. (2) Methods: We prepared the MeOH extracts of 107 different crude drugs, and screened their inhibitory effects on URAT1 by measuring the uptake of uric acid by HEK293/PDZK1 cells transiently transfected with URAT1. (3) Results: We found that the extract of the dried mature fruit of *Cnidium monnieri* inhibited urate uptake via URAT1. We isolated and identified osthol as the active ingredient from this extract. Osthol noncompetitively inhibited URAT1 with an IC_50_ of 78.8 µM. We evaluated the effects of other coumarins and found that the prenyl group, which binds at the 8-position of coumarins, plays an important role in the inhibition of URAT1. (4) Conclusions: *Cnidium monnieri* fruit may be useful for the treatment of hyperuricemia or gout in traditional medicine, and its active ingredient, osthol, is expected to be a leading compound for the development of new drugs for hyperuricemia.

## 1. Introduction

Hyperuricemia is a disease associated with lifestyle habits, and the increasing frequency of hyperuricemia and gout is most likely caused by westernized lifestyles and environments [1]. Hyperuricemia can be a risk factor not only for gouty arthritis and kidneys and ureteric stones, but also for cardiovascular and chronic kidney diseases [2,3]. Therefore, as an appropriate intervention, hyperuricemia should be tackled at an early stage [4].

Since humans have a nonfunctional urate oxidase gene, the blood concentration of uric acid, the end product of purine catabolism, is higher in humans than in other animals. In humans, the uric acid fraction that comes from foods or is synthesized from nucleosides is primarily excreted into the urine via glomerular filtration; however, most uric acid is reabsorbed by the proximal tubules via transporters. Another part of uric acid is excreted into the intestine via excretion-type transporters from the intestinal epithelial cells. Recent studies revealed that changes in the expression or functional changes in uric acid transporters are associated with hyperuricemia [5].

Urate transporter 1 (URAT1, SLC22A12) is expressed on the brush border membrane of proximal tubular cells, and is identified as the transporter that reabsorbs uric acid from the primary urine into the cells [6]. Apart from URAT1, glucose transporter 9 (GLUT9, SLC2A9), breast-cancer resistance protein ABCG2, organic anion transporter 1 and 3 (OAT1/3, SLC22A6/SLC22A8), multidrug resistance protein 4 (MRP4, ABCC4), and sodium phosphate transporter 4 (NPT4, SLC17A3) were identified as uric acid transporters, and these transporters play roles in the excretion of uric acid in the kidney and intestine [7,8,9,10].

Benzbromarone is a uricosuric agent and the first-line drug for the treatment of renal underexcretion-type hyperuricemia [1]. Benzbromarone increases the elimination of uric acid by the inhibition of URAT1 and uric acid reabsorption [6,11]. However, its usage is limited by its severe adverse effect in fulminant hepatitis [12]; thus, the discovery of safer URAT1 inhibitors is needed.

In this study, we screened new URAT1 inhibitors among 107 crude drugs used as components of formulations used in traditional Japanese Kampo medicine or as folk medicines. We identified osthol, the active ingredient of the fruit of *Cnidium monnieri*, as a promising lead compound for further drug-discovery efforts.

## 2. Results

### 2.1. Screening of 107 Crude Drug Extracts

In this study, we screened the inhibitory effects of 107 crude drug extracts on urate uptake via URAT1, and the results are presented in Figure 1. The crude drugs whose extracts exhibited an inhibitory effect on urate uptake with less than 50% of control at 100 µg/mL were Alpiniae Officinari Rhizoma, Amomi Semen, Araliae Cordatae Rhizoma, Arctii Fructus, Artemisiae Capillaris Flos, Atractylodis Rhizoma, Cimicifugae Rhizoma, Cinnamomi Cortex, Cnidii Monnieris Fructus, Cyperi Rhizoma, Schizonepetae Spica and Zingiberis Rhizoma Processum. Then, the cytotoxicities of these extracts were evaluated using the MTT assay. Finally, we observed that the extracts of Artemisiae Capillaris Flos, Cinnamomi Cortex, Cnidii Monnieris Fructus and Schizonepetae Spica exhibited an inhibitory effect of urate uptake with less than 50% of control without significant cytotoxicities at 100 µg/mL.

### 2.2. Effect of Cnidii Monnieris Fructus Extract on URAT1, and Its Activity-Guided Fractionation

Among these four crude drugs, we chose Cnidii Monnieris Fructus for further evaluation since it exhibited the highest inhibitory effect on urate uptake via URAT1. The MeOH extract of Cnidii Monnieris Fructus inhibited urate uptake via URAT1 in a concentration-dependent manner with the half maximal inhibitory concentration (IC_50_) of 53.2 µg/mL (Figure 2a). Cnidii Monnieris Fructus extract also exhibited a concentration-dependent cytotoxicity; however, cytotoxicity was not statistically significant at concentrations below 100 µg/mL (Figure 2b).

We tried to identify the active compound from Cnidii Monnieris Fructus. When the MeOH extract of Cnidii Monnieris Fructus was partitioned into four fractions, both hexane and AcOEt fractions exhibited significant inhibitory effects at concentrations equivalent to 100 µg/mL of the original extract (Figure 3). Since thin-layer chromatography (TLC) patterns of these two fractions contained the same spot, and since stronger signals were found for the hexane fraction than for the AcOEt fraction, the hexane fraction was further fractionated by open silica-gel chromatography into fraction A–N. Fractions D–H exhibited significant inhibitory effects on urate uptake via URAT1 at 33 µg/mL (D, 39%; E, 35%; F, 34%; G, 33%, H, 52%). These fractions had a spot with identical *Rf* value in the patterns observed by TLC, and osthol was collected from this spot by preparative TLC and identified by the spectra of ^1^H- and ^13^C- nuclear magnetic resonance (NMR), electron ionization-mass spectrometry (EI-MS) spectra. Moreover, the same elution time was observed by high-performance liquid chromatography (HPLC) analysis when using the standard compound. The chemical structure of osthol is shown in Appendix A.

### 2.3. Inhibitory Effect of Osthol on URAT1 and Its Kinetics

Osthol inhibited urate uptake via URAT1 in a concentration-dependent manner, and its IC_50_ value was 78.8 µM (Figure 4a). Osthol exhibited a concentration-dependent cytotoxicity, but it was not significant at the concentrations below 100 µM (Figure 4b). The concentration of osthol in the MeOH extract of Cnidii Monnieris Fructus was 28.3 (*w*/*w*) % as determined by HPLC analysis.

Kinetic analysis of the inhibitory effects of osthol on URAT1 was performed. The uptake of urate (25, 50, 100, and 200 µM) was measured with or without 100 µM of osthol for 1 min, and a Lineweaver–Burk plot was plotted (Figure 5). Two regression lines were crossed on the X-axis.

### 2.4. Comparison of Inhibitory Effects on URAT1 among Coumarins

The inhibitory effects of coumarins (Appendix A) were compared. Although coumarins, which have a basic structure, did not exhibit clear inhibitory effects, osthol and osthenol, which have an 8-prenyl group, exhibited a significant inhibitory effect on URAT1 at 100 µM. Bergaptol and bergamottin, which belong to furanocoumarins, and geraniol, which is a basic compound containing a prenyl group, did not inhibit URAT1 (Table 1).

## 3. Discussion

URAT1 exists at the brush border membrane of renal proximal tubular cells and reabsorbs uric acid from the primary urine into the blood circulation. It can be regarded as a pre-eminent target in drug discovery, as observed for the previous efforts that led to the discovery of enhancers of uric acid elimination, such as benzbromarone, probenecid [6], and lesinurad [13]. The activity of uric acid transportation via URAT1 in vitro can be evaluated by a *Xenopus laevis* oocyte induced with URAT1 or by vesicles of renal brush border membrane [6,14]. In HEK293 cells, the dual transfection of URAT1 and its anchor protein PDZK1 exhibited higher uptake of uric acid than the single transfection of URAT1 [15]. In the present study, we screened URAT1 inhibitors from 107 crude drugs used in traditional Japanese Kampo medicines and as folk medicines using HEK293/PDZK1 cells transiently transfected with URAT1, and found that the extract of Cnidii Monnieris Fructus and its active ingredient, osthol, significantly inhibited URAT1.

Cnidii Monnieris Fructus is originated from the dried mature fruit of *Cnidium monnieri* (Apiaceae), and it is used to disperse cold, dispel wind, dry dampness, warm the kidneys, fortify the yang, kill parasites, and stop itching in traditional Chinese medicine [16], and to treat skin sores, tinea, and itching as external medication in traditional Japanese Kampo medicine [17]. An experimental pharmacological study revealed that its extract exhibited immunostimulant [18], anti-inflammatory [19], antiallergic, antipruritic [20,21], and antiosteoporotic effects [22]. However, there are no reports of the usages of Cnidii Monnieris Fructus for hyperuricemia or gout in traditional medicines. Since Cnidii Monnieris Fructus (Shechuangzi in Chinese) is an approved drug in China [23], it is expected that this crude drug can also be used for the treatment of hyperuricemia or gout.

In the present study, the IC_50_ values of Cnidii Monnieris Fructus extract and osthol for urate uptake via URAT1 were 53.2 and 19.2 µg/mL (=28.5 µM), respectively. Since the extract at the IC_50_ value contained 15.1 µg/mL of osthol, osthol was responsible for 79% of the inhibitory effect of the extract. It is suggested that osthol could be considered the major active ingredient of Cnidii Monnieris Fructus extract for the inhibition of URAT1, and can be used as the marker compound for the quality control of enhancers of uric acid elimination.

Among natural products, it is reported that morin, one of the flavonoids isolated from the twigs of *Morus alba*, competitively inhibits URAT1 [24]. In this study, we did not evaluate the twigs of *M. alba*; instead, we evaluated the root bark of *M. alba* because the twig of this plant is not used as a crude drug in Japan. Chang et al. reported that EtOH extracts of the twigs or root bark of *M. alba* contained 21 mg/g of morin in both [25]. Although the MeOH extract of root bark of *M. alba* (Mori Cortex) in this study contained morin at some extent, in our screening experiment, we did not observe an inhibitory effect of *M. alba* root bark extract on URAT1. On the contrary, not only did *M. alba* root bark extract but some crude drug extracts stimulate the uptake of uric acid into the cells. Although the detailed mechanisms by which extracts increase uric acid uptake are unknown, we cannot exclude their possible effects on transporters other than URAT1, or on the expression and/or translocation of URAT1, since uptake studies did not focus on the initial phase of the uptake, but were performed for a relatively longer period (30 min).

We evaluated the inhibitory effects of coumarins on URAT1, and found that 8-prenyl coumarins, including osthol and osthenol, significantly inhibited URAT1. Since bergamottin, which is a 5-*O*-prenyl coumarin, and the simple prenyl compound geraniol did not inhibit URAT1, we postulated that the prenyl group taht binds at the 8-position of coumarins would play an important role in the inhibition of URAT1. Osthol was the noncompetitive inhibitor of URAT1 in the present study. Since uric acid is a heterocyclic compound without any prenyl groups, osthol could not compete with uric acid; instead, it could bind to an allosteric pocket of URAT1 to inhibit its function. Among coumarin derivatives, esculetin and fraxetin lowered the serum concentration of uric acid after oral administration in murine hyperuricemia models induced by oxonate [26]. However, esculetin and fraxetin did not inhibit URAT1 in the present study. Since esculetin and fraxetin restored the expressions of transporters of uric acid in the kidney, such as OAT1 and ABCG2 [24], osthol might act on these transporters to improve hyperuricemia, since osthol presents the basic chemical structure of coumarins, similar to esculetin and fraxetin.

Osthol was recently reported to show anticancer [27], cardioprotective [28,29,30], antihypertensive [29], anti-inflammatory, and antiasthma [31] properties, to suppress ulcerous colitis [32], to promote bone-fracture healing [33], to induce osteogenesis in osteoblasts [34], to protect chronic [35] and acute kidney failure [36], and to inhibit intimal hyperplasia [37]. In the present study, osthol exhibited a URAT1-inhibitory effect (IC_50_ = 78.8 µM) compatible with that of probenecid and indomethacin, whose IC_50_ values were 42 and 41 µM, respectively [11]. Though the IC_50_ value of osthol was larger than that of benzbromarone (less than 0.1 µM) [6] and lesinrad (3.5 µM) [38], there are no reports for *Cnidii Monnieris Fructus* causing liver injury, and osthol could not have the adverse effect of liver injury [12]. Therefore, we suggest that *Cnidii Monnieris Fructus* may be useful for the treatment of hyperuricemia or gout in traditional medicine, and that osthol can be used as a leading compound for drug discovery for hyperuricemia and gout. Further studies to validate osthol activity in vivo are needed.

## 4. Materials and Methods

### 4.1. Crude Drugs

We chose 107 crude drugs that are frequently used as ingredients in traditional Japanese Kampo formulations or as folk medicines in Japan, and purchased the crude drugs that met the grade standards of the Japanese Pharmacopoeia (17th Edition) [39] or of nonpharmacopoeial crude drugs (2015) [40] from several distributers (Appendix A). Five grams of each crude drug was sonicated in 100 mL of MeOH for 30 min and filtered. The residue was further extracted twice in the same manner, the three filtrates were merged, evaporated under reduced pressure, and finally lyophilized to yield the final crude extract. The Latin names, origins, lot numbers, and the extraction ratios yielded from each crude drug are shown in Appendix A. Each extract was suspended in dimethylsulfoxide at a concentration of 100 mg/mL, and kept at –20 °C until use.

### 4.2. Inhibitory Effect of the Samples on URAT1

HEK293 cells stably expressing the myc construct of PDZK1 (HEK293/PDZK1 cells) were previously obtained [41]. HEK293/PDZK1 cells were maintained in Dulbecco’s modified Eagle’s medium (Sigma-Aldrich, St. Louis, MO, USA) containing 10% fetal bovine serum (FBS, Invitrogen, Carlsbad, CA, USA) and 1 mg/mL G418 disulfate (Nacalai Tesque, Kyoto, Japan) in a humidified incubator at 37 °C under a 5% CO_2_ atmosphere. The cells were seeded in poly-L-lysine-coated 24-well plates (2.5 × 10^5^ cells/well) and incubated for 24 h. cDNA-encoding human URAT1 was subcloned into the pCMV–SPORT6 vector (Invitrogen). The constructs were transfected into HEK293/PDZK1 cells using a cationic liposome prepared with Hilly Max reagent (Dojindo Laboratories, Kumamoto, Japan) according to the manufacturer’s protocol. For mock cells, the pCMV–SPORT6 vector was transfected into the cells. Then, the cells were further incubated for 24 h. The transfected cells were preincubated for 7.5 min at 37 °C with 0.2 mL of chloride-free Hanks’ balanced salt solution (Cl^−^-free HBSS; 125 mM gluconic acid sodium salt, 4.8 mM potassium gluconate, 1.3 mM calcium gluconate, 1.2 mM KH_2_PO_4_, 1.2 mM MgSO_4_, 5.6 mM glucose, pH 7.4). Cl^−^-free HBSS containing 0.625 μCi/mL (=11.6 μM = 0.2 mg/dL) [8-^14^C] uric acid (American Radiolabeled Chemicals, St. Louis, MO, USA) with or without the samples were prepared, and the cells were incubated at 37 °C with 0.25 mL of these solutions for 30 min. Benzbromarone (Sigma) was used as positive control. The final concentration of DMSO in the medium was 0.1% or less. For kinetic analysis, [8-^14^C] uric acid was diluted with cold uric acid to a suitable concentration, and the incubation time was set at 1 min. The cells were washed three times with ice-cold PBS, and then 0.2 mL of 1 M NaOH was added. The plates were incubated overnight under gentle shaking, and the cell lysates were neutralized with 0.2 mL of 1 M HCl. An aliquot (20 µL) of the solution was used for protein analysis, and the remaining solutions were transferred into vials containing 4 mL of Clearzol I (Nacalai). Radioactivities were measured using a liquid scintillation counter. Protein concentrations were measured using a BCA^TM^ Protein Assay kit (Thermo Scientific, Rockford, IL, USA). The uptake amount of uric acid into the cells was compensated by the amount of protein, and the percentage of the control for uric acid uptake by URAT1 was calculated as follows:

Uptake of uric acid (% of control) = ((uptake into sample-treated URAT1-transfected cells) − (uptake into mock cells))/((uptake into control URAT1-transfected cells) − (uptake into mock cells)) × 100.

IC_50_ was calculated from the least-square regression line made from three points that crossed 50% of the control logarithmic concentration values.

### 4.3. Sample Cytotoxicity

HEK293/PDZK1 cells were seeded in 96-well plates (5 × 10^4^ cells/well) and incubated for 24 h. Then, the cells were incubated with Cl-free HBSS containing the sample for 30 min. The solutions were aspirated, and the cells were further incubated with a medium containing 10% FBS for 2 h. Then, the cells were incubated with a medium containing 3-(4,5-dimethylthiazol-2-yl)-2,5-diphenyl tetrazolium bromide (0.5 mg/mL; MTT, Sigma), and further incubated for 4 h. Following this, 10% sodium lauryl sulfate solution was added to the wells, and the plate was incubated for 20 h to dissolve formazan crystals produced by the cells. Optical density at 570 nm (OD_570_) was measured, and cell viability was calculated as follows:

Cell viability (% of control) = ((OD_570_ of sample-treated cells) − (OD_570_ of the background))/((OD_570_ of control cells) − (OD_570_ of the background)) × 100.

### 4.4. Preparation and Fractionation of Cnidii Monnieris Fructus Extract

Cnidii Monnieris Fructus (100 g; Lot #23610001, Tochimoto Tenkaido, Osaka, Japan) was sonicated in 0.7 L of MeOH for 30 min and filtered. The remaining residue was further twice extracted with MeOH; the resulting filtrates were combined, evaporated under reduced pressure, and lyophilized to yield the crude extract. This protocol was repeated four times, each with 100 g of *Cnidii Monnieris Fructus.* Altogether, 26.1 g of MeOH extract was obtained from 500 g of *Cnidii Monnieris Fructus*. This extract was suspended in 1.05 L of H_2_O, and was extracted with 450 mL of hexane (three times) to yield the hexane fraction. The water layer was extracted again with 450 mL of AcOEt (three times), and then extracted with 450 mL of water-saturated BuOH (three times). The hexane, AcOEt, BuOH, and water fractions yielded 6.1, 7.4, 2.3, and 2.5 g of crude extract, respectively.

The hexane crude fraction (3.0 g) was applied to open silica-gel chromatography (BW-200, Fuji Silysia, Kasugai, Japan; 5.5 i.d. × 22.5 cm) eluted with hexane:acetone (8:2). Factions A (79 mg), B (379 mg), C (211 mg), D (536 mg), E (44 mg), F (581 mg), G (865 mg), H (460 mg), I (250 mg), J (83 mg), K (107 mg), L (75 mg), M (140 mg), and N (130 mg) were collected from the crude mixture following the pattern observed by TLC (Appendix A). Fraction F (150 mg) was applied to preparative TLC (PLC silica gel 60 F_254_, 0.5 mm, Merck, Darmstadt, Germany) developed with hexane:acetone (6:4). The spot with *Rf* value 0.5 detected by the absorbance at 254 nm was collected to yield compound **1** (12 mg). The ^1^H- and ^13^C-NMR), and EI-MS spectra of compound **1** matched the ones previously described for osthol (Basile et al., 2009). The standard compound of osthol was purchased from LKT Laboratories Inc. (St. Paul, MN, USA; Lot #: 23927906), and HPLC analysis (column, Cosmosil 5C_18_-ARII, 4.6 × 150 mm, Nacalai Tesque, Kyoto, Japan; Solvent A (0.1% HCOOH)/B (MeOH) 70/30–0/100 (0–20 min), 1.0 mL/min, linear gradient; detection, UV at 322 nm) was conducted. compound **1** and osthol standard were eluted at 17.2 min, and the mixed solution of compound **1** and osthol standard exhibited a single peak. From these results above, compound **1** was identified as osthol.

Osthol (0.100, 0.200, and 0.400 µg, respectively) and Cnidii Monnieris Fructus MeOH extract (1.00 µg) were applied to HPLC analysis following the protocol described above, and the concentration of osthol in the extract was calculated by the absolute calibration line (*r*^2^ > 0.999) obtained from the peak areas of osthol solution.

### 4.5. Coumarins

Standard compounds of coumarins were purchased as follows: coumarin (Nacalai), 7-methoxycoumarin (Tokyo Chemical Industry, Tokyo, Japan), 4-methoxycoumarin (Wako Pure Chemicals, Osaka, Japan), 6-methoxycoumarin (Wako), umbelliferone (Wako), daphnetin (Wako), esculetin (Wako), fraxetin (Sigma), bergaptol (Extrasynthese, Geray, France), bergamottin (ChromaDex, Irvine, CA, USA), osthenol (ChemFaces, Wuhan, China), and geraniol (Kanto Chemical Industry, Tokyo, Japan).

### 4.6. Statistical Analysis

Statistical analysis was carried out by one-way analysis of variance (ANOVA) and Dunnett’s multiple comparison *t*-test using PASW Statistics software (version 18, SPSS; IBM, Armonk, NY, USA). A probability value of less than 0.05 was considered statistically significant.

## Figures and Tables

**Figure 1 molecules-23-02837-f001:**
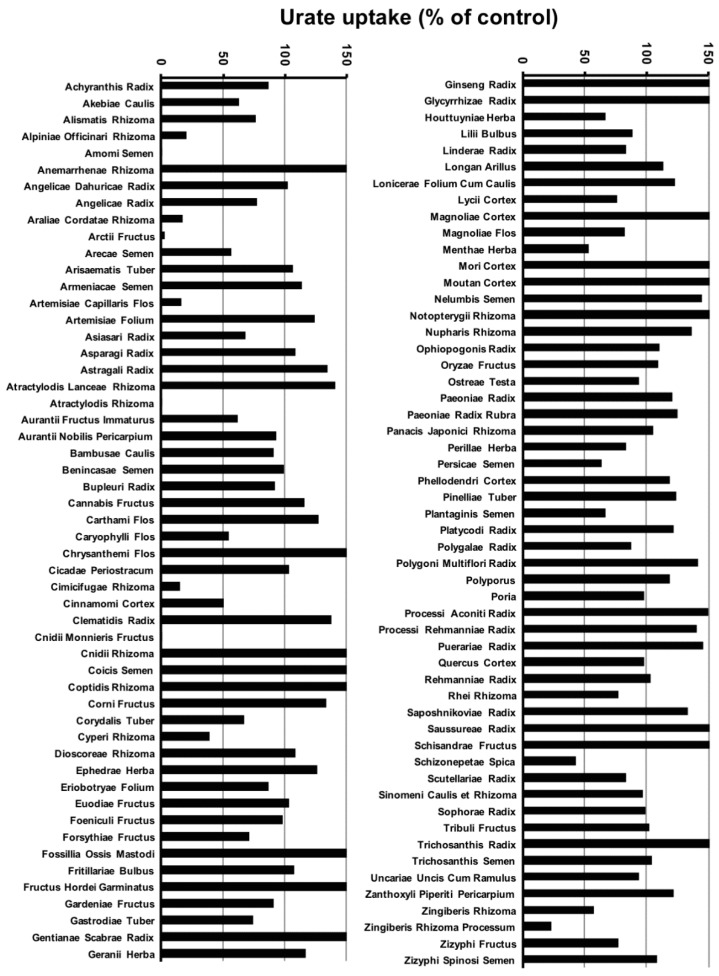
Effects of crude drug extracts on the uptake of uric acid via urate transporter 1 (URAT1). HEK293/PDZK1 cells were transiently transfected with human URAT1. The cells were incubated with uric acid (11.6 µM) with or without each crude drug extract (100 µg/mL) at 37 °C for 30 min, and the uptakes of uric acid into the cells were measured. The origins of crude drugs are listed in Appendix A. Data are expressed as % of control, calculated as described in Materials and Methods, and they represent averages of the experiments (*n* = 2).

**Figure 2 molecules-23-02837-f002:**
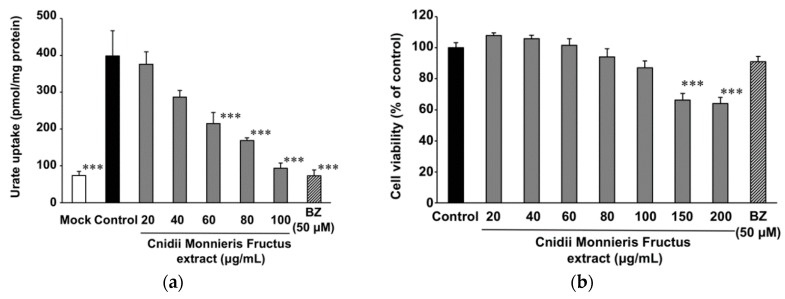
Effect of Cnidii Monnieris Fructus extract on the uptake of uric acid via URAT1. (**a**) HEK293/PDZK1 cells were transiently transfected with human URAT1. Cells were incubated with uric acid (11.6 µM) with or without the extract at 37 °C for 30 min, and the uptakes of uric acid into the cells were measured. Data are expressed as the mean ± S.E. (*n* = 3). (**b**) Cytotoxicity of Cnidii Monnieris Fructus extract was measured using the MTT method. Data are expressed as the mean ± S.E. (*n* = 6). Benzbromarone (BZ, 50 µM) was used as a positive control. *** *p* < 0.001 vs. the control group by analysis of variance (ANOVA) and Bonferroni–Dunnett’s multiple *t*-test.

**Figure 3 molecules-23-02837-f003:**
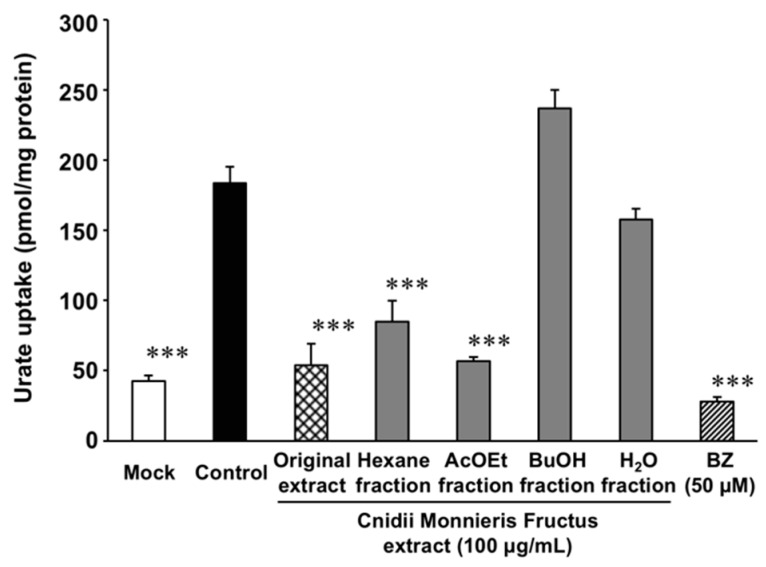
Effect of Cnidii Monnieris Fructus extract and its fractions on the uptake of uric acid via URAT1. HEK293/PDZK1 cells were transiently transfected with human URAT1. Cells were incubated with uric acid (11.6 µM) with or without the extract and its fractions at the concentrations related to Cnidii Monnieris Fructus extract (100 μg/mL), respectively, at 37 °C for 30 min, and the uptakes of uric acid into the cells were measured. Data are expressed as the mean ± S.E. (*n* = 3). 50 µM of BZ was used as a positive control. *** *p* < 0.001 vs. the control group by ANOVA and Bonferroni–Dunnett’s multiple *t*-test.

**Figure 4 molecules-23-02837-f004:**
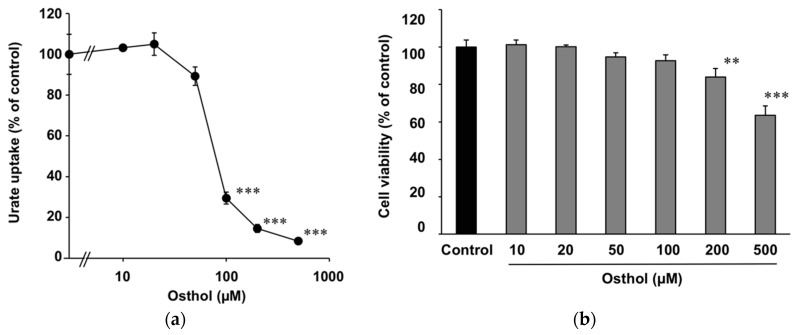
Effect of osthol on the uptake of uric acid via URAT1. (**a**) HEK293/PDZK1 cells were transiently transfected with human URAT1. Cells were incubated with uric acid (11.6 µM) with or without osthol at 37 °C for 30 min, and the uptakes of uric acid into the cells were measured. Data are expressed as the mean ± S.E. (*n* = 3). (**b**) Cytotoxicity of osthol was measured using the MTT method. Data are expressed as the mean ± S.E. (*n* = 6). ** *p* < 0.01 and *** *p* < 0.001 vs. the control group by ANOVA and Bonferroni–Dunnett’s multiple *t*-test.

**Figure 5 molecules-23-02837-f005:**
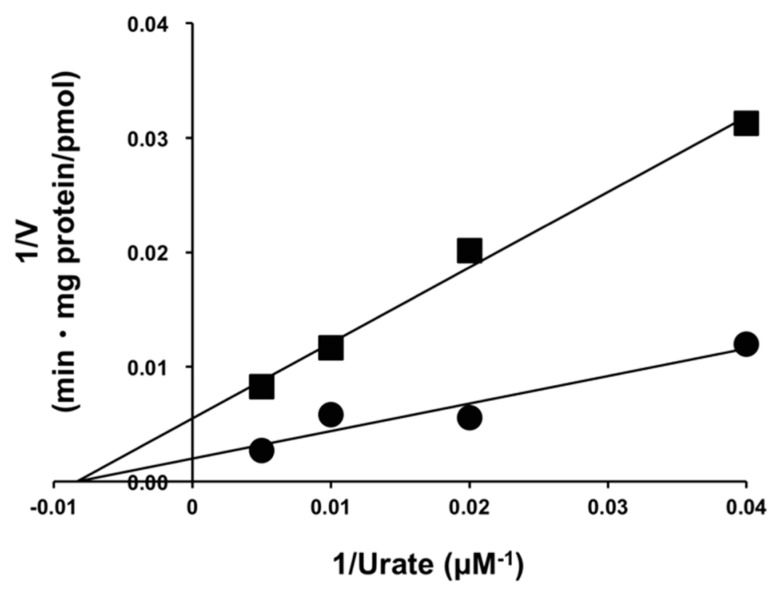
Kinetics analysis for the inhibitory effect of osthol on the uptake of uric acid via URAT1. HEK293/PDZK1 cells were transiently transfected with human URAT1. The cells were incubated with uric acid (25, 50, 100, and 200 µM) with (closed square) or without (closed circle) osthol (100 µM) at 37 °C for 1 min. The uptakes of uric acid into the cells (V) were measured, and a Lineweaver–Burk plot was made. Each symbol is expressed as the mean of quadruplicate data. Linear regression lines calculated using the least-square method are shown.

**Table 1 molecules-23-02837-t001:** Inhibitory effect of the related compounds of osthol on the uptake of urate via URAT1.

Compound	Urate Uptake (% of Control)
coumarin	111 ± 8
7-methoxycoumarin	137 ± 8
4-hydroxycoumarin	87 ± 9
6-hydroxycoumarin	83 ± 14
umbelliferone	114 ± 6
daphnetin	140 ± 10
esculetin	120 ± 5
fraxetin	136 ± 7
osthol	36 ± 10
osthenol	31 ± 4
bergaptol	83 ± 6
bergamottin	117 ± 6
geraniol	88 ± 5

HEK293/PDZK1 cells were transiently transfected with human URAT1, and incubated with 0.625 μCi/ml [8-^14^C] uric acid with or without each compound (100 μM) for 30 min. Then, the uptakes of radioactivity into the cells were measured. Data are expressed as mean ± S.E. (*n* = 3–4). Data are expressed as % of control calculated as described in the Materials and Methods.

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
