# Peer review of "Effects of Osthol Isolated from Cnidium monnieri Fruit on Urate Transporter 1"

_molecules, 2018, doi:10.3390/molecules23112837_

Reviewer 1 Report

It is well designed elegant work, supported by evidences.

What is the rationale for 11.6 uM of uric acid? Please cite reference for the same.

Authors are recommended to show the image of TLC pattern as well as silica gel chromatography, preparative TLC. It would be interesting to see the TLC pattern since the hexane fraction has been fractionated - Hexane has low polar index and it is difficult to separate the hexane fractions!

Fig 4 - Authors are requested to calculate the equivalent ug/ml or ng/ml values for osthol uM dose. Readers will be curious to know, whether the purification/fractionation of the crude extract results in the dilution of biological effect? This calculation will address those. And, these  findings should be discussed in the discussion.

What is the concentration of DMSO used for invitro experiments?

Author Response

Q1. It is well designed elegant work, supported by evidences.

What is the rationale for 11.6 uM of uric acid? Please cite reference for the same.

A1. Thank you very mych for reviewing our article. American Radiolabeled Chemicals sold [8-14C] uric acid 0.5 mCi/ml (= 9.28 mM) solution, and we determined the suiteble concentration of uric acid at 1:800 dilution of the original solution by the suitable dpm count of uptake study in a preliminary experiments. Uric acid 11.6 µM is about 0.2 mg/dL. This concentration is around 10% standard blood concentration of uric acid, and it is considered reasonable. We add “11.6 µM (= 0.2 mg/dL) at line 236. 

Q2. Authors are recommended to show the image of TLC pattern as well as silica gel chromatography, preparative TLC. It would be interesting to see the TLC pattern since the hexane fraction has been fractionated - Hexane has low polar index and it is difficult to separate the hexane fractions!

A2. We add the photo of TLC in Supplementary Fig. 2, and added the text at line 279.

Q3. Fig 4 - Authors are requested to calculate the equivalent ug/ml or ng/ml values for osthol uM dose. Readers will be curious to know, whether the purification/fractionation of the crude extract results in the dilution of biological effect? This calculation will address those. And, these  findings should be discussed in the discussion.

A3. We already show that “the IC50 values of osthol for urate uptake via URAT1 was and 19.2 µg/ml (= 28.5 µM)”, and discussed that since the extract at the IC50 value contained 15.1 µg/ml of osthol, osthol was responsible for 79% of the inhibitory effect of the extract by the comparison of both IC50s in line in line 168-171.

Q4. What is the concentration of DMSO used for in vitro experiments?

A4. We add the concentration of DMSO in line 239.

Reviewer 2 Report

This paper is really interesting since it discuss new possible ways to target an important problem such as hyperuticemia. Only two thins need to be added before acceptance:

- When speaking about the cardiovasculsr risk of hyperuricemia two new important paper should be cited: redon p et al, j hyp 2018 (epub ahead of print) and maloberti a et al, j clin hyp 2018, 20: 193-200.

- In the results and conclusion differences from brenvomarone were discussed. Also differences between the founded molecule and probenecid and lenisurad need to be discussed.

Author Response

Q1. This paper is really interesting since it discuss new possible ways to target an important problem such as hyperuticemia. Only two thins need to be added before acceptance:

- When speaking about the cardiovasculsr risk of hyperuricemia two new important paper should be cited: redon p et al, j hyp 2018 (epub ahead of print) and maloberti a et al, j clin hyp 2018, 20: 193-200.

A1. Thanks for valuable comments. We add these articles as references at line 38.

Q2. In the results and conclusion differences from brenvomarone were discussed. Also differences between the founded molecule and probenecid and lenisurad need to be discussed.

A2. According to the reviewer’s comments, we add the discussion at line 204-207.

Reviewer 3 Report

Ref No: Molecules-379317

Effects of osthol isolated from Cnidium monnieri fruit on urate transporter 1

Journal: Molecules

1.      I reviewed the manuscript by Yuusuke Tashiro et al., entitled “Effects of osthol isolated from Cnidium monnieri fruit on urate transporter 1”. The English language need more improvement before publication.

2.      There are several typo mistakes in the front page of title.

3.      “Cnidium monnieri” is the scientific name must be italic glyphs in the text.

4.      In results section: suggestions add segment title (or sub title). For example, 2.1 Effect of Cnidii Monnieris Fructus extract on the uptake of uric acid via URAT1. And add 2.2, 2.3……

5.      Please add “Conclusions” clearer and more specific integration.

Author Response

Q1. I reviewed the manuscript by Yuusuke Tashiro et al., entitled “Effects of osthol isolated from Cnidium monnieri fruit on urate transporter 1”. The English language need more improvement before publication.

A1. This article has already been taken by the native speaker checking. We add the certificate of English checking. And five days for the revision are not enough for the additional English checking.

Q2. There are several typo mistakes in the front page of title.

A2. In Abstract, we find some typo mistakes, and we correct in the revised article. Thank you for checking.

Q3. “Cnidium monnieri” is the scientific name must be italic glyphs in the text.

A3. We found this in Abstract, and correct it with Italic letters. Thank you for checking.

Q4. In results section: suggestions add segment title (or sub title). For example, 2.1 Effect of Cnidii Monnieris Fructus extract on the uptake of uric acid via URAT1. And add 2.2, 2.3……

A4. Thank you for your comment. We carefully considered your comment, we add the sub-section titles in Result section. 

Q5. Please add “Conclusions” clearer and more specific integration.

A5. Thank you for your comment. We carefully considered your comment, and add one sentence at 203-210.
